# In Vitro Diffuse Large B-Cell Lymphoma Cell Line Models as Tools to Investigate Novel Immunotherapeutic Strategies

**DOI:** 10.3390/cancers15010235

**Published:** 2022-12-30

**Authors:** Matylda Kubacz, Aleksandra Kusowska, Magdalena Winiarska, Małgorzata Bobrowicz

**Affiliations:** 1Department of Immunology, Medical University of Warsaw, 02-097 Warsaw, Poland; 2Doctoral School, Medical University of Warsaw, 02-091 Warsaw, Poland; 3Laboratory of Immunology, Mossakowski Medical Research Institute, Polish Academy of Sciences, 02-106 Warsaw, Poland

**Keywords:** diffuse large B-cell lymphoma, cell line models, monoclonal antibodies, antibody-drug conjugates, CAR-T cell therapy

## Abstract

**Simple Summary:**

Despite a range of emerging immunotherapeutic strategies, the aggressive nature of DLBCL poses an ongoing clinical challenge. Therefore, there is a need to pursue more effective treatment modalities based on monoclonal antibodies, antibody-drug conjugates and CAR-T cell therapy. Up to now, the above-mentioned immunotherapeutic options have been extensively studied with the aid of established cell line models, which have significantly facilitated proceeding from preclinical to clinical investigations and led to improvement of DLBCL treatment. However, there are still several important challenges associated with faithful recapitulation of the aggressive nature of DLBCL. Therefore, the current review discusses means in which cell line models fulfill an essential tool leading to greater understanding DLBCL biology and development of novel immunotherapeutic strategies.

**Abstract:**

Despite the high incidence of diffuse large B-cell lymphoma (DLBCL), its management constitutes an ongoing challenge. The most common DLBCL variants include activated B-cell (ABC) and germinal center B-cell-like (GCB) subtypes including DLBCL with MYC and BCL2/BCL6 rearrangements which vary among each other with sensitivity to standard rituximab (RTX)-based chemoimmunotherapy regimens and lead to distinct clinical outcomes. However, as first line therapies lead to resistance/relapse (r/r) in about half of treated patients, there is an unmet clinical need to identify novel therapeutic strategies tailored for these patients. In particular, immunotherapy constitutes an attractive option largely explored in preclinical and clinical studies. Patient-derived cell lines that model primary tumor are indispensable tools that facilitate preclinical research. The current review provides an overview of available DLBCL cell line models and their utility in designing novel immunotherapeutic strategies.

## 1. Introduction

The most common non-Hodgkin lymphoma (NHL) subtype, diffuse large B-cell lymphoma (DLBCL), accounts for up to 40% of all lymphoid tumors [1]. Its management remains an ongoing challenge with up to 50% of patients relapsing or developing resistance (r/r patients) [2]. The treatment is most often based on a combination of rituximab (RTX) with cyclophosphamide, vincristine, doxorubicin and prednisone (R-CHOP) [3]. To this date, several factors behind R-CHOP resistance have been identified, with high genetic and clinical heterogeneity as one of the most significant [4].

In 2000, Alizadeh et al. identified two main subtypes: germinal center B-cell (GCB)-like and activated B-cell (ABC)-like subtype, with the latter being characterized with significantly worse prognosis [5]. Discovery of several unique DLBCL subtypes based on cell of origin (COO) as well as molecular features has complemented International Prognostic Index (IPI) in identifying high-risk disease and predicting therapy efficacy [3]. However, there is no universally accepted way in integrating various prognostic factors in DLBCL [6].

A subgroup of DLBCL with MYC, BCL2 and BCL6 rearrangement can be identified which is linked to worse prognosis [7,8]. Development of DLBCL may additionally occur as a life-threatening complication, named as Richter transformation (Richter syndrome, RT) [9]. RT is development of an aggressive lymphoma on the background of chronic lymphocytic leukemia (CLL) and occurs in approximately 2–10% CLL’s patients [10,11]. High DLBCL heterogeneity (reviewed by Yanguas-Casás et al. [12]) is reflected at the molecular level by DLBCL model cell lines [12]. The cell lines are relatively easy to harvest so they can be used at almost any moment to analyze their features in flow cytometry and colorimetric or enzymatic assays [13]. Those advantages make cell line models a useful tool for investigation of novel therapies, which have recently heralded a new era in DLBCL management [14].

## 2. Established Cell Lines 2-Dimenstional (2D) Models as Tools to Study DLBCL Biology

### 2.1. Cell Line Establishement from Primary Cell Culture

Since establishment of the first leukemia-lymphoma (LL) cell lines in 1964 [15] a vast number of cellular models have been characterized [16]. Factors behind frequent failure in cell line establishment are varied and not fully understood; however, it has been reported that cells derived from r/r patients with unfavorable prognostic features possess enhanced growth potential in vitro, indicating a higher success rate for cell line establishment [17]. Isolation and propagation of the cells are illustrated in Figure 1 and thoroughly described in [18,19].

### 2.2. Challenges in Establishment of Cell Lines Faithfully Recapitulating DLBCL

Immortalized LL cell lines offer a repeatable and reliable diagnostic tool to model primary tumor in the clinical arena around the world [20]. However, using non-authenticated cell lines is associated with a risk of about 1:6 for choosing a false, contaminated cell line [21]. In 1999, when up to 14.8% of human hematopoietic lines were false cell cultures due to cross-contamination, Drexler et al. raised awareness on a recurring problem of overgrowth of cell lines by other cellular models and mycoplasma contamination [21]. Furthermore, a percentage of the established cell lines, the so-called B-lymphoblastoid cell lines (referred to as EBV+ B-LCLs), resulted from unintended immortalization of non-malignant B cells by “passenger” EBV. EBV+ B-LCLs carry genetic aberrations, which may mimic malignancy-associated features and lead to misidentification as malignant cells [16,21]. Authentication testing is of utmost importance to ensure a panel of non-contaminated, high-quality cell line models with stabile phenotypic characteristics [22]. In depth molecular characterization and validation of leukemia and lymphoma cell lines in LL-100 panel provided valuable data on the genetic variability of these cancer models facilitating the understanding of LL pathogenesis [23].

Noteworthy, cell line models often arise from patients with end-stage, non-nodal, and leukemic phase lymphomas that possess a wide-ranging repertoire of biased mutational sequences [24]. Therefore, Caesar et al. optimized a strategy to purify and facilitate ex vivo expansion of non-malignant human B cells at germinal center stage, that can be genetically modified to enable combinatorial expression of putative tumor suppressor genes [24,25].

It is also common to perform experiments using cell line models representative of other B-cell malignancies and extrapolating them to DLBCL as certain features overlap between them [26,27]. The most frequently used ones are BL cell lines [28], since BL tumors double in size in approximately 25 h [29] and therefore the established BL models possess a high growth potential, which makes them relatively easy to harvest and sustain in vitro. A panel of BL lines includes Raji, the first continuous line of hematopoietic origin [15], Daudi, Ramos and BJAB [28]. FL and/or MCL lines are also frequently used [28].

### 2.3. DLBCL Cell Line Models Available to Study Biology of the Aggressive B-Cell Lymphoma

DLBCL biology is studied with highly heterogeneous models representative of specific subtypes [28]. Cell line models established in two research centers: Stanford University (SU) and Ontario Cancer Institute (OCI) have dominated DLBCL research [30,31,32]. Epstein et al. established three EBV-negative SU-DHL cell lines (SU-DHL-1, -2 from pleural effusion; SU-DHL-3 from peritoneal effusion), followed by seven next ones (from pleural effusion: SU-DHL-7, SU-DHL-8, SU-DHL-9, and SU-DHL-10; from peritoneal effusion: SU-DHL-4, SU-DHL-6; from a lymph node: SU-DHL-5), which comprise a group of highly diverse malignancies arising from B lymphocytes [30]. Phenotypic characterization and histological description of EBV-negative cell lines of B-lineage including OCI-Ly-1, OCI-Ly-2, OCI-Ly-3, OCI-Ly-4, OCI-Ly-7, OCI-Ly-8, and OCI-Ly-18 have been reported by Tweeddale et al. [31] and Chang et al. [32]. They were established either at diagnosis (OCI-4, OCI-8, OCI-18) or during a patient’s relapse (OCI-1, OCI-2, OCI-3, OCI-7). Interestingly, OCL-Ly-3 serves as an example of cell line dependent on growth factors as it undergoes self-regulation by IL-6 [33].

#### 2.3.1. Panel of ABC-DLBCL Models

ABC-DLBCL, which constitutes around 30% of DLBCL cases, is characterized with a clinically unfavorable outcome as compared to GCB-DLBCL [34]. Several ABC-DLBCL representing cell lines, including Ri-1 (Riva), SU-DHL-2, SU-DHL-9, OCI-Ly18, HBL-1, RC-K8, U-2946, or TDM8 are available.

Unique features of the available cell lines allow for the choice of tailored preclinical models. For instance, RC-K8 with dysregulated Rel/NF-κB pathway is an appealing candidate for testing novel immunotherapeutic agents for ABC subtype [35]. Moreover, ULA serves as a model of multidrug resistance as it originated from the patient who had resisted several chemotherapy courses [26]. Lastly, U-2946 is an example of a cell line with MCL1 (member of BCL2 family) overexpression, which is a recurrent feature in ABC-DLBCL that promotes drug resistance and overall cancer cell survival [36].

#### 2.3.2. Panel of GCB-DLBCL Models

GCB-DLBCL constitutes up to 50% of DLBCL cases and has a GC phenotype defined as CD10+, BCL6+ [34]. Although its management has been more successful than of the ABC subtype, the high genetic heterogeneity can lead to unexpected clinical outcome in a number of patients [6,34,37]. Cell line models of the GCB subtype include RL, SU-DHL-4, SU-DHL-6, SU-DHL-8, SU-DHL-10, OCI-Ly1, OCI-Ly3, OCI-Ly7 or Karpas422. 

Interestingly, Karpas422, bearing both t(14;18) and t(4;11) along with several other abnormalities, is an attractive model to study chemotherapy-resistant NHL as it originated from DLBCL patient unresponsive to consecutive chemotherapeutic schemes [38].

One of the hallmarks of GCB-DLBCL, present in up to 30% of cases, is t(14;18) translocation, which results in Bcl-2 overexpression [39,40]. Additionally, t(14;18) in GCB-DLBCL is associated with significantly worse prognosis compared to GC-type DLBCL without the translocation (29 to 63% 2-year survival, respectively) [40]. GCB-DLBCL cell line models containing t(14;18) include for example SU-DHL4, SU-DHL6, and OCI-Ly1.

#### 2.3.3. Cell Line Models with Unique Features Representative of RS-DLBCL and Secondary DLBCL

Unique cell line representatives of RS-DLBCL formed on CLL background and secondary DLBCL formed on Hodgkin lymphoma (HL) background have also been developed. RS-DLBCL represents a group of mainly CD5+ and CD23+ large blast-like neoplastic B cells with small nucleoli with diffuse growth pattern and high proliferative potential [41]. Up to 95% of RS-DLBCL cases represent the more aggressive ABC-subtype with a variable BCL6 expression, MUM1+ and CD10- [42]. Despite its unique features, current recommendations for RS management are the same as for aggressive NHLs or de novo DLBCL [43]. Importantly, high expression of programmed death (PD-1) and programmed death ligand (PD-L1) observed in RS patients [44] make them appealing targets for immunotherapy (reviewed by Iannello et al. [45]).

Generally, aggressive NHL has a worse prognosis in comparison to HL [46,47]. However, survival of HL patients with a relapse, especially after chemotherapy, is comparable to patients with primary aggressive NHL [48,49]. The first heterogenic HL-NHL cell line, established by Amini et al. [50], U-2932, was further shown to be composed of two different subpopulations with unique phenotypic features [51]. Additionally, the expression levels of CD20 and CD38 antigens in both populations were demonstrated to change through the 100 days of culturing, raising questions about the stability of the U-2932 model, and hence its reliability [52]. Furthermore, Sambade at al established a cell line with chromosomal rearrangements similar to recurrent aberrations occurring in both HL and NHL (including microsatellite instability), recognized as U-2940 [53]. Initially, U-2940 was reported to arise from DLBCL [53]; however, further research ascribed it as a model of primary mediastinal B-cell lymphoma (PMBL) [54,55].

Table 1 provides a concise summary of thoroughly characterized DLBCL cell lines.

## 3. 3-Dimensional (3D) Models

Despite being an indispensable tool within cancer research, most of the cell lines fail to reliably recapitulate the importance of genetic and microenvironmental factors of tumors in disease progression [65]. Thus, 3-dimensional (3D) culture systems that provide an insight into pathophysiology of tumor microenvironment and allow to monitor cell functions (such as proliferation, differentiation, motility, and metabolism) [66] offer a testbed for therapeutic agents prior to in vivo studies [67].

Duś-Szachniewicz et al. established 3D spheroid models based on Ri-1 and Raji cells, which were used to test cytotoxicity of doxorubicin (DOX) and ibrutinib (IBR) on B-NHLs [68]. Co-culturing of lymphoma cells with stromal cells led to a reduced IBR-induced apoptosis in comparison to the 3D monoculture, which recapitulates the significance of stromal cells in tumor pathophysiology. Lara et al. investigated the effect of different RTX isotypes generated by recombinant DNA technology on 2D and 3D-cultured B-cell lymphoma lines (Raji, Daudi, BJAB and Granta-519) and observed considerable differences in potency of RTX-mediated complement-dependent cytotoxicity (CDC) with respect to antibody isotype and model structure, with 3D models limiting penetration of RTX and limiting its cytotoxic activity [69]. The promising results encourage further investments in the generation of faithful 3D model systems investigating other B-cell malignancies, including DLBCL.

## 4. Significance of Cell Line Models Resistant to Therapeutic Agents

Years of RTX employment as well as recent findings on mechanisms of insensitivity to a range of immunotherapies indicate that developing resistance is an unavoidable side effect of the regimens [70]. Generating cellular models of acquired resistance is based on repeated incubations of cells with increasing concentrations of a cytotoxic agent (Figure 2) [71].

RTX-resistant cell lines (RRCLs) were generated by Czuczman et al. from Raji, SU-DHL-4, RL and U-2932 [72]. The cells acquired resistance mainly via CDC, due to medium supplementation with human serum, and additionally to antibody dependent cell-mediated cytotoxicity (ADCC) [72]. Additionally, CD20 underwent downregulation and expression of pro-apoptotic members of the BCL-2 family (Bax and Bak) was reduced [72,73]. RRCLs hold potential to be used in investigations of immunotherapeutic strategies that aim to overcome the RTX resistance phenomenon.

The mechanisms of potential insensitivity to CD37-targeting agents can also be investigated in cell line models. Accordingly, Arribas et al. generated two DLBCL cell line models (SU-DHL-2 and SU-DHL-4), which were resistant to IMGN529/DEBIO1562 (an anti-CD37 ADC) and demonstrated different phenotypic changes within the models [74]. SU-DHL-2 (ABC-DLBCL model) carried CD37 loss and 25 mutations in kinases or transcription factors, SU-DHL-4 sustained CD37 expression but carried 48 shared mutations in genes encoding for cytokines, kinases, oncogenes, and transcription factors. Melhus et al. presented work exploring expression characteristics and intrinsic factors associated with sensitivity of 55 lymphoma cell lines (including 20 GCB subtype and 7 ABC subtype) to 177^Lu-lilotomab satetraxetan, anti-CD37 radioimmunoconjugate [75]. Intrinsic treatment resistance to 177^Lu-lilotomab satetraxetan is evident; however, only in a subset of cell lines and is independent of genetic lymphoma hallmarks including TP53, BCL2, and MYC.

## 5. Investigating Immunotherapeutic Strategies

### 5.1. Exploring Potential of Monoclonal Antibodies (mAbs)

The availability of various lymphoma cellular models is vital for testing novel antibodies in preclinical trials. Several B-cell antigens e.g., CD19, CD22, CD37, CD40, CD47, CD79b, and CD80 have been proposed as targets for mAbs mainly for r/r patients [76].

Gehlert et al. used B-ALL cell lines (SEM, Jurkat, CEM, MOLT-16, and Nalm-6 cells) to test efficacy of CD19-targeting mAb, tafasitamab [77]. The mAb was optimized in a way to improve antibody hexamerization, which enhanced CDC but had no influence on antibody-dependent cellular phagocytosis (ADCP) or ADCC. Additionally, combination of tafasitamab and RTX improved cytotoxicity against B-ALL cell lines in vitro [78]. Although the study did not include DLBCL cell lines, preclinical studies performed on B-ALL models encouraged investigation of tafasitamab efficacy (in monotherapy or in combination with lenalidomide) in r/r DLBCL patients who cannot be qualified for ASCT.

As demonstrated in a range of preclinical studies with cellular models, the CD37 antigen is widely expressed across multiple types of B-cell lymphoid neoplasms [79], which prompted extensive development of CD37-targeting mAbs, as recently reviewed [80]. Lately, a panel of DLBCL models (OCI-Ly7, OCI-Ly19, RC-K8, Ri-1, SU-DHL-4, SU-DHL-8, WSU-DLCL-2, and U-2932) were used to confirm efficacy of bispecific CD37 antibody (DuoHexaBody-CD37), which triggered potent ADCC, ADCP and superior CDC to other tested CD37-targeting mAbs [81].

Since it has been shown that CD38 is an important prognostic marker also in DLBCL [82], preclinical studies evaluating efficacy of anti-CD38 mAbs like daratumumab in DLBCL are undertaken. In in vitro and in vivo models of DLBCL (cell lines: Toledo, WSU-DLC2, SU-DHL-4, SU-DHL-6), MCL and FL potent daratumumab-mediated ADCC and ADCP was demonstrated independently of CD38 expression [83].

Furthermore, Bouwstra et al. demonstrated that high expression of CD47, the so-called “don’t eat me” immune checkpoint, correlated with detrimental effect on OS in non-GCB DLBCL patients after R-CHOP therapy [84]. Further studies on DLBCL cell lines (OCI-Ly3, U-2932, SU-DHL-2, SU-DHL-4, SU-DHL-6, SU-DHL-10) demonstrated increased therapeutic effect of RTX after blocking CD47 in non-GCB DLBCL model [84]. Moreover, B-NHL co-cultures (DLBCL lines: Pfeiffer and Karpas422; BL; Raji, Daudi; FL: RL) showed that ADCC and ADCP induced by a CD47- and CD19-targeting bispecific antibody (TG-1801) was enhanced when it was used in a “U2-regimen” (with anti-CD20-mAb: ubilituximab; and PI3Kδ/CK1e inhibitor: umbralisib) than in monotherapy [85].

By now, it has been discovered that PD-L1 is aberrantly expressed on HLs and little is known about its role in NHLs [86]. Analysis performed on a panel of several NHL cell lines including 28 DLBCL models revealed that PD-L1 expression was confined to only 3 models (HBL-1, OCI-Ly-10 and RC-K8) [87]. Astonishingly, as shown with the aid of various human DLBCL cell lines (including OCI-Ly-3, TDM8, SU-DHL-4), the PD-L1 levels can be upregulated by vincristine administration improving efficacy of the PD-L1 blockade therapy [88]. An extensive testing of immune checkpoint inhibitors in preclinical investigations have inspired clinical trials that eventually led to FDA’s approval of nivolumab and pembrolizumab for certain types or r/r lymphomas [89,90].

### 5.2. Antibody-Drug Conjugates (ADCs) and Targeted-Drug Delivery

Antibody-drug conjugates (ADCs) are monoclonal antibodies bound to cytotoxic payload that is delivered directly to the tumor cells [91]. Immunologic functions of ADC such as CDC and ADCP are weakened to strengthen the antitumor effect of the molecule, which induces killing of the cells [92].

Approval of polatuzumab vedotin (Pola) in 2019, CD79b-targeting ADC, benefited a significant percentage of r/r DLBCL patients [93]. Nonetheless, insensitivity to ADCs may appear as in other immunotherapeutic options. DLBCL cell line panel consisting of Pola-sensitive (DB, STR-428, SU-DHL10, SU-DHL-4, NU-DUL-1, U-2932) and Pola-resistant (SU-DHL-8, HT, SU-DHL-2, RC-K8) cell lines served as a tool to investigate resistance mechanisms to Pola, which included low CD79b expression, high expression of anti-apoptotic Bcl-xL and ABC transporters [94]. Interestingly, exposition of SU-DHL-8, SU-DHL-2 and HT (DLBCL cell lines) resistant to anti-CD79b ADC (Pola) leads to CD20 upregulation and enhances RTX sensitivity via CDC and ADCC [95]. This justifies the use of combination therapy for Pola with RTX and bendamustine regimen.

Moreover, sensitivity of 27 commonly used DLBCL cell lines of ABC and GCB subtypes to MMAE-conjugated ADCs: anti-CD22 (pinatuzumab vedotin) and anti-CD79B ADCs (Pola) was assessed [96]. Although majority of the cells were sensitive to both ADCs, Farage, HS445 and HT responded only to anti-CD22; OCI-Ly3, HBL-1, and Pfeifer models responded only to anti-CD79b. Only SU-DHL-2 and WSU-NHL were resistant to both drugs.

Furthermore, a potent in vitro activity against multiple B cell lines (including Ramos, Raji, Daudi, Farage, and RL) via ADCC, ADCP, and CDC was exerted by naratuximab emtansine (IMGN529), an investigational CD37-targeting ADC conjugated to DM1 [97]. The results were confirmed on a similar panel of DLBCL models (U-2932, SU-DHL-4, DOHH-2, OCI-Ly18, OCI-Ly7, and Farage) by Hicks et al. who further demonstrated increased apoptosis and cell death of those cellular models as a consequence of synergistic anti-tumor potency of naratuximab ematansine and anti-CD20 agents, especially RTX [75]. In a similar pattern, U-RT-1 cell line was used to demonstrate high antitumor activity of anti-CD37 alfa-amanitin-conjugated antibodies [98].

### 5.3. CAR-T Cell Therapy

Introduction of T cells modified with chimeric antigen receptors (CARs) has led to a significant advancement in management of multiple malignancies, including lymphoid neoplasms [99]. By now, three CAR-T cells (axicabtagene ciloleucel, lisocabtagene maraleucel and tisagenlecleucel) have been registered in treatment of r/r DLBCL. Surprisingly, majority of preclinical studies testing efficacy of CARs were performed on B-ALL cell lines, not on DLBCL models. However, the existing ones, provide important directions in optimizing CAR T-cells.

The great success of anti-CD19 CAR-T cells in B-ALL encouraged preclinical studies on their utility against DLBCL. Interestingly, improved killing of B-cell lymphoma lines, including OCI-Ly2 and OCI-Ly19, Raji, Daudi, DEL (anaplastic large cell lymphoma model), Granta-519 and Jeko-1 (MCL), was triggered by the combination of anti-CD19-CAR-T cells with an anti-CD20-IFN fusion protein indicating that antibody-targeted IFN could improve the CAR-T cell therapy [100]. Noteworthy, in Raji and U-2932 anti-CD19-CAR-T demonstrated higher efficacy when combined with ibrutinib than in monotherapy [101]. Although no pathway responsible for synergistic effect was identified [101], the experiment suggests that this approach might offer a benefit to B-cell lymphoma patients.

Furthermore, promising findings on anti-CD37 mAbs also encouraged exploration of CD37 as a target for CAR-T cells. The panel of several B-cell lymphoma models (BL-41, Daudi, Granta-519, K422, K562, Jeko-1, Jurkat, Maver-1, MINO, Raji, Ramos, ROS-50, SC-1, SU-DHL-6, SU-DHL-4, Oci-Ly3, Oci-Ly7, and Oci-Ly10) and two xenograft mice models were constructed to analyze the cytotoxic effect of CD37CAR [102].

Despite the promising results on CD19 CAR-T cells, around half of high-grade lymphoma patients develops resistance after receiving such treatment [103]. Unfortunately, resistance mechanisms to CAR-T cell therapy in DLBCL are not well-understood yet, as most of the observations arise primarily from ALL studies [70]. By now, there is also a lack of existing cellular models with developed insensitivity to CAR-T cell therapies that could be used to optimize therapeutic approaches.

Both Table 2 and Figure 3 provide a concise summary of utility of cellular models in exploring targeted immunotherapies for DLBCL. 

## 6. Xenograft Mouse Models

Animal models are a powerful tool to test novel immunotherapeutic options prior to clinical investigations [104]. In essence, xenograft mouse models of human lymphoma cells can be established by implantation of stable cell lines of primary tumor samples into immunosuppressed recipient or humanized mice. Serial passaging of the engrafted tumors may be necessary to achieve high xenotransplantation efficiency. Transplantation occurs into tail vein (disseminated model) or subcutaneously (orthotropic model). In case of disseminated models the transplanted cells are most often genetically modified to express luciferase and tumor growth is measured using bioluminescence [105]. In orthotropic models bioluminescence imaging and measurement by calipers can be applied [106]. By mimicking genetic alterations found in the human disease, the utilized models (e.g., genetically engineered mouse models (GEMMs) have allowed the detailed in vivo investigation of several lymphoma-associated oncogenes and tumor suppressors, shedding light on their role in normal B cell development and tumorigenesis. However, it has to be said that the following approach is associated with specific advantages and disadvantages; for instance GEMMs cannot reproduce the genetic complexity and the heterogeneity of the human tumors, an aspect especially important when aiming at discovery and pre-clinical testing of novel therapeutics. Additionally, a few other important challenges accompany the utility of xenograft models, including the lack of an immune response against the tumor and lack of physiological microenvironment [104,107]. Nonetheless, animal research is an indispensable step before proceeding to clinical trials.

Murine models are commonly utilized in testing efficacy and safety of mAbs, ADCs and CAR-T cells. Xenograft models derived from different cell lines injected subcutaneously (BL: Raji, Ramos, Namalwa; DLBCL: SU-DHL-6; acute lymphoblastic leukemia: SUP-B15) demonstrated significant lymphoma inhibition by XmAb5574, an anti-CD19 mAb with an Fc-engineered domain for effector function improvement. [108]. Furthermore, the growth of each NHL xenograft model, obtained by cell lines’ (BL: Raji, Ramos; MCL; FL and DLBCL cell lines) implantation above the right flank, was inhibited by highly cytotoxic anti-CD37 agent, recognized as AGS67E [109]. Importantly, murine models lessen efforts associated with designing strategies, which aim at overcoming RTX-resistance. A panel of human lymphoma cells (BL: Raji, Ramos; DLBCL: RL, U2932, SU-DHL-5; lymphoblastic lymphoma: U-698-M; RRCLs: Raji 2R, Raji 4RH, and RL 4RH; ofatumumab-exposed cell lines: Raji, U-2932, RL) inoculated via tail vein injection to generate respective xenograft models enabled comparison of efficacy of two anti-CD20 mAbs: RTX and ofatumumab [110]. In vivo studies, performed on xenografted tumors after subcutaneous injection of Pola-sensitive (e.g., SU-DHL-4, U-2932) and Pola-resistant (e.g., SU-DHL-2, RC-K8) cell lines, showed that Pola increased CD20 expression in Pola-resistant xenograft models and had an enhanced antitumor activity when combined with RTX [94]. Since extrapolating findings based on BL models to DLBCL is a widely accepted and justified practice, it comes as no surprise that numerous in vivo studies utilize Raji xenograft model for testing CAR-T efficacy. In particular, potency of CD19 CAR constructs, observed in both disseminated and orthotropic models [111], or synergistic effect of CD19 CAR combined with ibrutinib, analyzed with the aid of a disseminated model [112], have been thoroughly investigated in Raji-derived xenograft models. Studies on MCL disseminated xenograft models (Mino, JEKO-1) demonstrated additive effect of CD19 CAR combined with ibrutinib [113] and identified bispecific CD79b/CD19 [114]. CAR as strategies to improve standard-of-care therapies against MCL. In addition, profound efficacy of CD19 CAR-based therapies against B-ALL in clinical settings prompted the establishment of leukemic humanized mice with human immune system as a valuable pre-clinical model [115].

## 7. Conclusions

The current review has summarized available cell lines used as powerful tools in investigating DLBCL biology as well as in evaluating therapeutic efficacy of antitumor agents. Despite the wide arsenal of cellular DLBCL models representative of ABC and GCB subtypes, only several are repeatedly used, including SU-DHL-4, SU-DHL-6, OCI-Ly-3, TDM8, or U-2932, to optimize available immunotherapies. To ensure reliable modeling of the primary tumor it is crucial to analyze a genetic background of each model and ensure it remains uncontaminated with foreign material. Importantly, several lines have been authenticated and included in the LL-100 panel (ABC-DLBCL: NU-DHL-1, OCI-Ly3, Ri-1, U-2932, U-2946; GCB-DLBCL: DOHH-2, OCI-Ly7, OCI-LY19, SU-DHL-4, SU-DHL-6, and WSU-DLCL2).

Effective employment of cell line models encourages other research teams to utilize the same model systems, which on the one hand facilitates the reproducibility of the generated data among research centers yet limits the scope of investigations. After all, DLBCL remains one of the most heterogenic lymphoid malignancies and requires multiple cellular models to study its biology. Simultaneously, it is crucial to report failures associated with culturing of specific cell lines to allow for verification and perhaps modifications of the models to overcome culturing challenges. Surprisingly, U-2932 remains a frequently used cell line in DLBCL research regardless of the existing controversies on its complex phenotype, which might perhaps serve as an advantage in development of more universal immunotherapeutics. Furthermore, it remains a common strategy to use other B-cell lymphoma models, especially BT cell lines (Raji, Daudi, Ramos) to yield results and extrapolate them to DLBCL.

Importantly, the wide arsenal of the existing cellular models contributed significantly to the acceleration of the preclinical testing of various immunotherapeutics, which has been observed in the past decade along with approval of a range of mAbs, ADCs, and CAR T. However, years of immunotherapy employment demonstrated that it inevitably leads to development of resistance, which remains an ongoing challenge to overcome. Consequently, an urge to expand even wider arsenal of commonly available patient-derived cell lines and to develop agent-resistant cell lines persists. Furthermore, investment in 3D systems is required to provide insight into efficacy of anti-tumor molecules before proceeding with animal testing. The attempts aiming at optimizing modeling of the primary tumor will benefit in improved optimization of (immuno)therapeutic strategies.

## Figures and Tables

**Figure 1 cancers-15-00235-f001:**
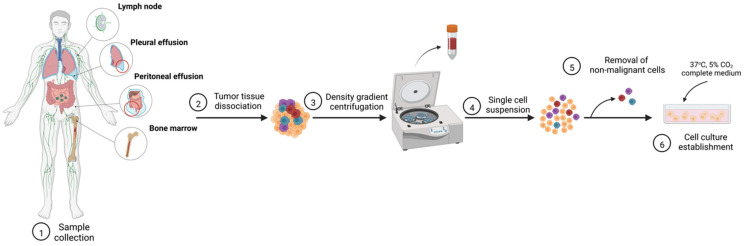
Establishment of patient-derived cell line models. Created with BioRender.com, accessed on 30 October 2022.

**Figure 2 cancers-15-00235-f002:**
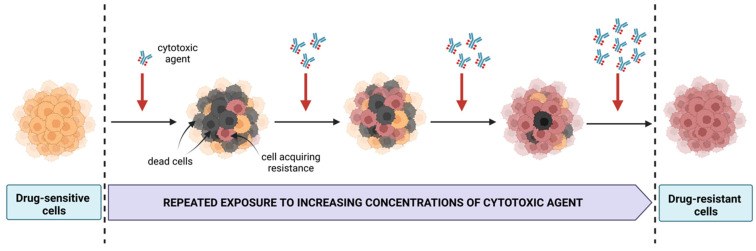
Generation of cell line models resistant to therapeutic agents. Created with BioRender.com, accessed on 30 October 2022.

**Figure 3 cancers-15-00235-f003:**
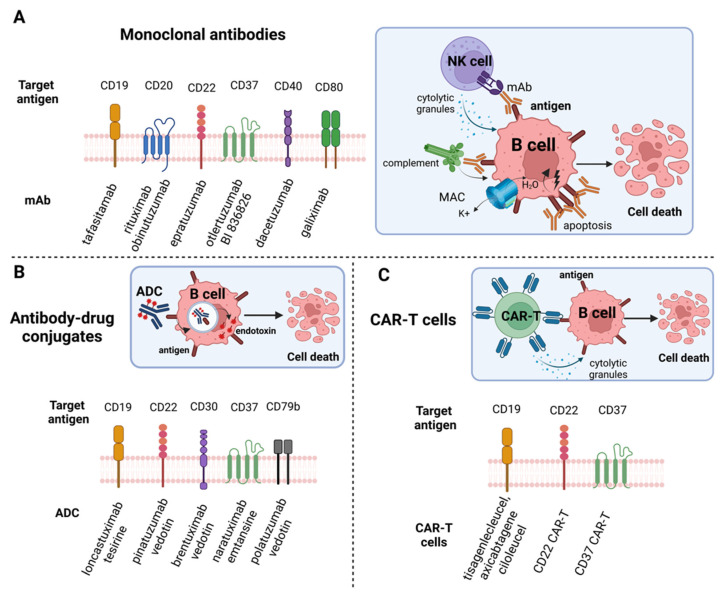
Targeted immunotherapies in DLBCL investigated on cellular models. (**A**) Monoclonal antibodies (mAbs); (**B**) Antibody-drug conjugates (ADCs); (**C**) chimeric antigen receptor T cells (CAR T-cells). Created with BioRender.com, accessed on 30 October 2022.

**Table 1 cancers-15-00235-t001:** Overview of the cell lines used to study DLBCL.

Cell Line	Citation	Age, Sex	Cell Source and Clinical Stage	EBV-Status	Immunophenotype	Characteristic Mutations, Aberrations or Translocations	Culture Requirements
ABC-DLBCL cell line models							
HBL-1	Abe et al. [56]	65 y.o. male	Pleural effusion	Negative	(IgM, *K),* Bl’, BA-I+, and HLA-DR’.	t(14;16)	RPMI-1640 No glutamine Doubling time: 48 h
HBL-2	Abe et al. [56]	84 y.o. male	Cervical lymph node biopsy	Negative	Monoclonal surface Igs (IgM, D, A), Fcy-receptors, C3 receptors, Bl’, and HLA-DR’. Weak positivity for IgD	t(11; 14)(q13,32)	RPMI-1640 Glutamine Doubling time: 48 h
ULA	Berglund et al. [57]	57 y.o. male	Ascites IV stage	Negative	CD10+, CD19+, CD20+, CD22+, CD27+, CD38+, CD40+, CD79b+, IgM+, IgD+, lambda light chain+, FMC7+, BCL-2+, BCL-6+, MHC class I and II+	p53-gene loss (14;18)(q32;q21)	Opti-MEM (45%) IMDM (45%) Glutamine Could not survive in RPMI-1640 Doubling time: N/A
TDM8	Tohda et al. [58]	62 y.o male	Bone marrow	Negative	CD5+, CD19+, CD20+, HLA-DR+, s-IgM, s-kappa	48, XY, add(1)(p3?), add(1)(q42), add(6)(p2?), del(6)(q?), +9, i(9)(p10) × 2, 15p, +18, −19, +mar karyotype	α-MEM No glutamine Doubling time: 30 h
OCI-Ly18	Chang et al. [32]	56 y.o male	Pleural effusion, High grade DLBCL	Negative	CD19+, CD20+, CD21+, CD23+, CD34+	Translocations involving bands 8q24 14q32 18q21	IMDM No glutamine Doubling time: 36 h
RC-K8	Kubonishi et al. [59]	55 y.o male	Peritoneal effusion	Negative	Complement receptors+, Ia+, B1+, and Leu 12 antigens+	14q+ chromosome, EBNA- t(11;14)(q23;q32)	RPMI-1640 No glutamine Doubling time: 48–60 h
U-2946	Quentmeier et al. [36]	52 y.o male	Pleural effusion, IV stage DLBCL	Negative	CD20+, CD79a+, CD10+, BCL-6+, MYC+, p53+ Partial expression of MUM1 and FOXP1	t(8;14)	RPMI-1640 Glutamine Doubling time: 48 h
GCB-DLBCL cell line models							
Karpas422	Dyer et al. [38]	72 y.o female	Pleural effusion	Negative	CD19+, CD37+, IgM+, and IgG+, 30% of cells IgD+ Weak CD10+ Stable expression of CAMPATH-1 (CDw52) as in normal lymphocytes	t(14;18) t(4;11)	RPMI-1640 No glutamine Doubling time: 60–90 h
MYC, BCL-2/BCL-6 rearrangement models							
EJ-1	Goy et al. [60]	43 y.o female	Ascites, IV stage DLBCL	Negative	CD10+, CD19+, CD20+, CD22+, CD45+, CD79b, sIgM, and light chain lambda+	t(14;8) t(8;14) del(7)(q31q32)	RPMI-1640 Glutamine Doubling time: 24 h
RC	Pham et al. [61]	Unknown	Pleural effusion, High-grade DLBCL	Negative	CD10+, CD19+, CD20+ (a small subset), CD22+, CD23+, CD38+, CD43+, CD44 (only partially), CD45+, CD79b+	t(2;8)(p12;q24.2) t(14;18)(q32;q21.3)	RPMI-1640 No glutamine Optimally could be split 1:2 every 3–4 days.
U-2973	Boström et al. [62]	42 y.o male	Peripheral blood mononucleated cells at diagnosis	Negative	CD19+, CD20+, CD22+, CD10+, CD38+, cytoplasmic CD79a, and dim kappa surface Ig. FMC7+ (only partially), CD52+ CMYC+, and BCL2+	t(14;18)(q32;q)	RPMI-1640 No glutamine Doubling time: 34 h
Models or Richter’s transformation							
U-RT1	Schmid et al. [63]	60 y.o male	Lymph node biopsy	Positive	CD20+, CD23+, BCL-2+, PAX-5+ CDKN2A- a chromosomal gain of the NOTCH1 gene locus	CLL cells: 13q14.3 17p13.1 (loss of a single copy of TP53) deletions as well as a mutation in the other TP53 copy (c.342-343del2bpins1bp) No TP53 loss in a lymph node biopsy material; however, a subset of cells still carried 13q14.3 deletion.	IMDM No glutamine Doubling time: approximately 36 h
VR09	Nichele et al. [64]	75 y.o male	Bone marrow sample	Positive	CD19+, CD20+, CD22+, CD23+, CD43+, CD45+, CD38+, CD138+, IgD+, IgM+, IgG+, kappa chain+, ZAP-70+ BCL-2+, MNDA+, and MUM1+	Chromosome 12 trisomy	RPMI-1640 No glutamine 10% DMSO Doubling time: 84 h
Models of secondary lymphomas							
U-2932	Amini et al. [50]	29 y.o female	Pleural effusion Nodular sclerosis HL type 1 with progression to ABC-DLBCL	Negative	The Hodgkin and Reed–Sternberg (HRS)-cells of the HL: CD30+, CD45+, CD15+, LMP-1+, p53+, Rb+, BCL-2+, BCL-6+ DLBCL cells: CD20+, CD30+, CD45+, CD15+, LMP-1+, p53þ, Rb+, BCL-2þ, BCL-6þ U-2932 line: LMP-1+, p53+, Rb+, BCL-2+, and BCL-6+	R1: upregulation of BCL2 and BCL6 R2: upregulation of BCL2 and MYC	RPMI-1640 Glutamine Doubling time: N/A

**Table 2 cancers-15-00235-t002:** Overview of immunotherapeutic strategies and cell lines used in evaluation of their efficacy.

Immunotherapy	Cell Lines	Reference
Monoclonal Antibodies (mAbs)		
Tafasitamab (anti-CD19 mAb)	B-ALL cell lines: SEM, Jurkat, CEM, MOLT-16, Nalm-6 cells	[78]
DuoHexaBody-CD37	DLBCL cell lines: OCI-Ly7, OCI-Ly19, RC-K8, Ri-1, SU-DHL-4, SU-DHL-8, WSU-DLCL-2, U-2932	[81]
Daratumumab (anti-CD38 mAb)	DLBCL cell lines: Toledo, WSU-DLC2, SU-DHL-4, SU-DHL-6 MCL cell lines: Jeko, REC-1, Mino, UPN1 FL cell lines: SC-1, WSU-FSCCL BL cell lines: Daudi	[83]
Anti-CD47 in combination with RTX	DLBCL cell lines: OCI-Ly3, U-2932, SU-DHL-2, SU-DHL-4, SU-DHL-6, SU-DHL-10	[84]
Anti-PD-L1 in combination with vincristine	DLBCL cell lines: OCI-Ly-3, TDM8, SU-DHL-4	[88]
Antibody-drug conjugates (ADCs)		
Polatuzumab vedotin (anti-CD79b ADC)	DLBCL cell lines: Pola-sensitive: DB, STR-428, SU-DHL10, SU-DHL-4, NU-DUL-1, U-2932 Pola-resistant: SU-DHL-8, HT, SU-DHL-2, RC-K8	[94]
Pinatuzumab vedotin (anti-CD22 ADC) versus Polatuzumab vedotin (anti-CD79b ADC)	DLBCL cell lines: U-2932, RIVA, TDM8, OCI-Ly10, OCI-Ly3, HBL1, BJAB, Pfeiffer, Farage, SU-DHL-6, SU-DHL-10	[96]
Naratuximab emtansine (Anti-CD37 ADC)	DLBCL cell lines: Farage, RL BL cell lines: Ramos, Raji, Daudi	[74]
Naratuximab emtansine (Anti-CD37 ADC) with RTX	DLBCL cell lines: U-2932, SU-DHL-4, DOHH-2, OCI-Ly18, OCI-Ly7, Farage	[75]
Anti-CD37 alfa-amanitin conjugated ADCs	DLBCL cell line: U-RT-1	[98]
CAR-T cell therapy		
Anti-CD19 CAR-T cells with anti-CD20-IFN fusion protein	DLBCL cell lines: OCI-Ly2 and OCI-Ly19 BL cell lines: Raji, Daudi Anaplastic large cell lymphoma cell line: DEL MCL cell lines: Granta-519, Jeko-1	[100]
Anti-CD37 CAR-T cells	DLBCL cell lines: SU-DHL-6, SU-DHL-4, Oci-Ly3, Oci-Ly7, Oci-Ly10, K422 MCL cell lines: Granta-519, Jeko-1, MINO, Maver-1, FL cell lines: SC-1 BL cell lines: Daudi, Raji, Ramos, BL-41	[102]

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
