# Peer review of "In Vitro Diffuse Large B-Cell Lymphoma Cell Line Models as Tools to Investigate Novel Immunotherapeutic Strategies"

_cancers, 2022, doi:10.3390/cancers15010235_

Round 1
Reviewer 1 Report
In vitro diffuse large B-cell lymphoma cell line models as tools to investigate novel immuno-2 therapeutic strategies
No: 2107059
Journal: Cancers MDPI
Article Type: Review
The current review paper by Matylda Kubacz and colleagues provides a good overview of the available DLBCL cell line models. I enjoyed reading this paper and believe that it will be of general interest to the readers of the journal. In addition, it is very useful for those working with DLBCL cell cultures and targeted immunotherapies such as monoclonal antibodies, conjugates, and CAR cells.
The paper is well-written, and I have only minor suggestions.
I wish you all the best. Kind regards.
Comments
Please, insert the references in the lines below: 29, 32, 33. Line 37, insert the reference for the study of 1963. Lines 63, 64, 74 (for each subtype); 83, 85, 116, 126, 141, 151, 161, 168, 170, 191, 192
Extra spaces: lines 45, 77
Table 1: I suggest not centralizing (vertical) the content of the cells. It is difficult to follow.
Author Response
Dear Reviewer,
I would like to thank you for the time you spent on reading our manuscript. We truly appreciate your warm words, thorough work, and very specific comments.
I hope that you will be satisfied with the changes we have introduced to the manuscript.
Warm wishes,
Matylda Kubacz
Reviewer 2 Report
In this review, the authors summarized the cell line models for DLBCL. This overview provides a valuable reference for researchers working on pre-clinical cell line studies with this disease.
Some minor issues:
1. Full names were not provided for quite some abbreviations, such as CLL (Line 31), CDC(Line 118), ADCP (Line 144), et al.
2. Some typos and editing errors were identified: 3D model (Line 109), ~" ADC Pola~" (Line 175), CD79b (Line 178), et al.
3. The authors described the utility of cell lines for immunotherapy/mAb drug tests. It would be helpful to readers if these pieces of information could be summarized in a new table or incorporated into Table 1.
4. Although in-vitro models are to be focused on in this review, some discussions with xenograft models derived from these cell lines will help.
Author Response
Dear Reviewer,
I would like to thank you for your time, which you spent on reading our manuscript. We appreciate all your suggestions, especially the ones that concerned enriching the contents of the manuscript. We believe that your comments significantly helped us convey the importance of the careful selection of cell line models for specific preclinical investigations. Furthermore, we believe that your idea of adding a new table and a section about xenograft models will make the current review more approachable to readers.
I hope that you will be satisfied with the changes we have introduced to the manuscript.
Kind regards,
Matylda Kubacz